



# Short-to-medium range hydrologic forecast to manage water and agricultural resources in India

Reepal Shah[1], Atul Kumar Sahai[2], Vimal Mishra[1]

[1]Civil Engineering, Indian Institute of Technology (IIT) Gandhinagar and ITRA Project: Measurement to Management (M2M): Improved Water Use Efficiency and Agricultural Productivity through Experimental Sensor Network.

[2]Indian Institute of Tropical Meteorology (IITM) Pune, India

*Correspondence to*: Vimal Mishra (vmishra@iitgn.ac.in)

**Abstract.** Water resources and agriculture are often affected by the weather anomalies in India resulting in a disproportionate damage. While short to medium range prediction systems and forecast products are available, a skilful hydrologic forecast of runoff and root-zone soil moisture that can provide timely information has been lacking in India. Using precipitation and air temperature forecasts from the Climate Forecast System v2 (CFSv2), Global Ensemble Forecast System (GEFSv2) and four products from Indian Institute of Tropical Meteorology (IITM), here we show that the IITM ensemble mean (mean of all four products from IITM) can be used operationally to provide hydrologic forecast in India at 7-45 days lead time. The IITM ensemble mean forecast was further improved using bias correction for precipitation and air temperature. Forecast based on the IITM-ensemble mean showed better skill in majority of India for all the lead times (7-45 days) in comparison to the other forecast products. Moreover, the VIC simulated forecast of runoff and soil moisture successfully captured the observed anomalies during the severe droughts years. The findings reported herein have strong implications for providing timely information that can help farmers and water managers in decision making in India.

## 1. Introduction

Droughts in India have enormous implications for water resources and agriculture (Mishra *et al* 2014, Shah and Mishra 2015). Many regions in India face drought risks due to lack of monsoon season rainfall. In 2015, a large part of India was under drought which affected agriculture and water resources (Times of India, 18 June 2016). Moreover, in 2015, about 33 million people were affected by the drought that covered 256 districts, 10 states, and caused an estimated loss of 650,000 crore Indian rupee (Indian Express, 11 May, 2016). Drought in 2015 had socio-economic effects on people as agriculture based industries were closed, water was supplied using trains, and at a few places farmers committed suicide. The major driver of hydrological (based on runoff) or agricultural (based on soil moisture) droughts in India remains the Indian summer monsoon (Mishra *et al* 2014, Shah and Mishra 2015),which accounts for about 80% of the mean annual rainfall and has 10% year-to-year variability (Rajeevan *et al* 2005, 2006, Rahman *et al* 2009). However, during the recent decades, increased air temperature has affected hydrologic and agricultural droughts in India (Mishra *et al* 2014) and in the other parts of the World (Dai *et al* 2004, Park Williams *et al* 2012, Shukla *et al* 2015, Livneh and Hoerling 2016).





One of the relatively well known reasons of droughts in India is the positive sea surface temperature anomaly (El Nino) in the Pacific Ocean (Kumar *et al* 1999, 2006) and in the Indian Ocean (Mishra *et al* 2012, Roxy *et al* 2015). However, in the

absence of hydrologic forecast at appropriate lead time, planning of agricultural and water resources sectors are often adversely affected. For instance, many times the cost of seeds, field preparation, and transplantation can not be recovered due to prolonged anomalies of soil moisture or rainfall. Furthermore, water resources, reservoir operations, and irrigation planning is affected in the absence of a skilful forecast at sufficient lead time. Prediction of anomalies in meteorological and hydrological conditions well in advance can assist timely decision-making to minimize impact on agricultural and water

resources sectors. Shah and Mishra (2016b) showed potential of the Global Ensemble Forecast System (GEFS; Hamill *et al* 2013) for hydrologic predictions in India with a lead time up to 7 days. They reported that up to 7-days lead time, major skill in hydrologic prediction is derived from initial hydrologic conditions (i.e. initial soil moisture content) as shown in Shukla and Lettenmaier (2011). Yuan et al.(2011) reported that soil moisture forecast from the CFSv2 (Saha *et al* 2014) provides useful information to predict droughts in the tropical region. Moreover, Yuan et al (2012a) showed that the CFSv2 can

provide a better seasonal hydroclimatic forecast than ensemble streamflow prediction in the USA.

Despite the utility of the various forecast products that can provide useful skill in hydrologic predictions, efforts have largely been limited to evaluate the potential of these products to provide forecast at 7-45 days lead time that can be used for agricultural and water resources planning in India. Here we provide an assessment of skill in hydrologic forecast at 7-45 days

lead time using data from GEFSv2, CFSv2, and IITM to improve management of water and agricultural resources in India.

## 2. Data and Methodology

### 2.1 Observed data

Forecast products were evaluated against observed data from India Meteorological Department (IMD). We used the 0.25°

daily gridded precipitation product from IMD which was developed based on ground observations from 6995 stations across India using an inverse distance weighing scheme (Shepard 1984) and is available for the period of 1901-2015 (Pai *et al* 2015). The IMD precipitation captures spatial variability of the monsoon season rainfall and features related to orographic rainfall in the Western Ghats and foothills of Himalaya. We used 0.5° daily observed maximum and minimum temperatures from IMD, which were developed based on 395 stations across India (Srivastava *et al* 2009). Gridded air temperature dataset

is available for the period of 1951-2013 and have been used in many previous studies (Mishra *et al* 2014, Shah and Mishra 2014, 2015, 2016).

### 2.2 Forecast Products





We evaluated prediction skill of precipitation, maximum and minimum temperatures from the CFSv2 reforecast (Saha *et al* 2014), GEFSv2 reforecast (Hamill *et al* 2013) and forecast products from IITM. Reforecast from the CFSv2 are based on a dynamical coupled model and are available at every 5[th] day from the start of year from the National Center of Environmental Prediction (NCEP). Moreover, 6-hourly forecasts at every 5[th] day are available with up to nine months lead time and at 1º

resolution for the period of 1982-2009 from the CFSv2. Climate forecast System (CFS) model's atmospheric component is operational at T126 spectral truncation (~100 Km horizontal resolution) and 64 sigma-pressure hybrid vertical resolution. Shukla and Lettenmaier (2011) using CFSv2 reported that initial hydrologic conditions dominate skill of hydrologic prediction in the continental United States (CONUS) up to 1-month lead time, beyond which skill from meteorological forcings dominated. McEvoy *et al* (2016) recently demonstrated higher skill for potential evapotranspiration than

precipitation using the CFSv2. Moreover, Yuan et al. (2011) reported that the CFSv2 performs better than CFSv1 for prediction of precipitation and air temperature in the United States. Mo and Lettenmaier (2014) found that for shorter lead times (about 1 month) the CFSv2 forecast has higher skill for soil moisture prediction than the benchmark forecast (climatological mean). Tian et al.(2016) evaluated the CFSv2 for the CONUS and found that extreme indices based on temperature were better predicted than that of precipitation.

Other than CFSv2, we compared precipitation and temperature forecast from the GEFSv2 reforecast (Hamill *et al* 2013), which is based on the Global Forecast System (GFS) model, for 7 and 15 days lead time. Ensemble members are generated in GEFS by making perturbations in initial atmospheric conditions which lead times to 11 ensemble members. The GEFS model runs at T254L42 resolution (~40 Km horizontal resolution) for the first 8 days lead time and at T190 (~54 km) for

lead time beyond 7.5 days. The GEFS reforecast are available at 1º resolution for lead time up to 16 days and at 0.5° for 8 days lead from 1985 to present. Shah and Mishra (2016b) evaluated skill of the GEFSv2 reforecast for drought prediction in India up to lead time 7 days and found that the GEFS reforecast showed correlation of more than 0.5 against drought estimates from the observed data.

We obtained four forecast products from IITM. The forecast products of IITM are generated from the same CFSv2 model described above. Abhilash et. al.(2014) have developed an ensemble prediction system using CFSv2 at T126 horizontal resolution (~100km) [called as IITM-CFST126] for prediction of monsoon intraseasonal oscillations (MISO) over Indian monsoon region 15-20 days in advance. They found that though the skill was reasonable, but there was significant dry bias over the Indian land. Sharmila et.al. (2013) reported that CFSv2 simulates the northward propagating MISO reasonably well

but it has cold bias in SST and tropospheric temperatures. Thus, Abhilash et al (2013) implemented a lead time dependent SST bias correction and forced the GFS (Atmospheric component of CFSv2) with slightly different physics and showed that it has improved skill over Indian Land compared to the CFSv2 (called as IITM-GFST126). Subsequently, Sahai et. al. (2015a) have also implemented a high resolution version of CFSv2 (at T382 horizontal resolution ~35km; called as IITM-CFST382) and showed that it has better skill in steep orographic regions. Although these three individual models show





similar prediction skill and their errors saturate at about the same lead time of around 25 days, there are many instances where the three models disagree in predicting particular events, such as the amplitude and phase of monsoon intraseasonal oscillation (MISO) propagation. Considering these facts, Abhilash et al (2015) have proposed a CFS based MME which improved the spreaderror relationship and adds value to both the deterministic and probabilistic forecasts. Real time skill for

these models has been reported in various communications. (Sahai *et al* 2013, Borah *et al* 2015, Joseph *et al* 2015b, 2015a, Sahai *et al* 2015b) Subsequently, bias corrected SST forced GFS was also run at T382 resolution (called as IITM-GFST382). Thus IITM's forecasts are available for four models, named IITM-CFST126, IITM-GFST126, IITM-CFST382, and IITM-GFST382, respectively. Model integrations for the year starting since 2001 to 2015 are carried out from 16th May and continued up to 28th September at every 5 day interval (16th May, 21st May, 26th May,..., 23rd Sep, 28th Sep) for the next

45 days period. Ensemble mean of all four IITM products (IITM-ensemble) and individual products were compared with the CFSv2 and GEFSv2 to evaluate the hydrologic prediction skill. The aim of this comparison was to evaluate if IITM forecast products provide better prediction skill than CFSv2 and GEFSv2. Moreover, the product that provide the best hydrologic prediction skill in India can be used operationally to forecast hydrologic conditions and rainfall and temperature anomalies that can help in decision making in agricultural and water resources.

We used ensemble mean (of all available ensemble members) of individual forecast products for evaluation. We selected forecasts at every 15[th] day, which was evaluated for 7, 15, 30, and 45 day lead time using accumulated precipitation and average temperature. We selected forecasts starting from 16[th] May till end of September as currently IITM provides forecast during the monsoon season. IITM will extend forecast to the non-monsoon season in near future. We aggregated all the

observed and forecast variables (precipitation, maximum and minimum air temperatures) to daily-scale (if they were available at sub-daily time period) and regridded them to 0.25° horizontal resolution to make them consistent with the spatial resolution of observed data. We regridded precipitation and air temperature using Maurer et al.(2002) which uses the Synergraphic Mapping System (SYMAP) algorithm (Shepard 1984) for precipitation and lapse rate based on elevation data for air temperature. We, however, carefully evaluated all the products at their original spatial resolution and at 0.25° to make

sure that datasets are consistent at both resolutions for spatial and temporal variability. We considered a common period of 2001-2009 for comparison and evaluation of different forecast products against the observed gridded data from IMD.

### 2.3 Forecast Evaluation

For evaluation of the forecast from each product against the observations, we prepared yearly time-series of precipitation and

temperature forecast for each forecast date by accumulating precipitation and averaging temperature for a given lead time (7-45 days). For instance, if the date of forecast was 1[st] June and lead time 15 days, accumulated precipitation and mean temperature for 15 days from June 1[st] for each of the products were estimated for the period 2001-2009. As the period for evaluation was 2001-2009, sample size was 10, and we acknowledge that a larger sample size with data for a longer




retrospective record will help us to better categorize uncertainty in forecast skill. We used coefficient of correlation, mean absolute error (MAE), and critical success index (CSI) to evaluate performance of the forecast products. A non-parametric Spearman Rank Correlation coefficient (Wilks 2006) was used to evaluate performance of forecast products in capturing temporal relationship with OBS. For this the forecast product and corresponding OBS are assigned ranks and then

correlation was estimated using following equation (2.1)

$$r_s = 1 - \frac{6 \sum d_i^2}{n(n^2 - 1)} \tag{2.1}$$

Where $r_s$ is Spearman Rank Correlation coefficient; $d_i$ is difference in rank between paired forecast and OBS; and $n$ is sample size (here 10). Significance of correlation was tested using the exact permutation distribution test (Robson 2002). Observed samples were permuted and rank correlations were estimated. Estimated correlation is significant if it

rejects the null hypothesis at 5% significance level.

Mean absolute error (MAE) was used to estimate error in the forecast products as compared to OBS. Absolute error was estimated in all the forecast products for each year as compared to OBS and then mean of all years was taken to estimate MAE. Critical success index (CSI; Wilks 2006) was used to evaluate anomalies predicted using forecast products as compared to OBS. CSI is ratio of hit (i.e. forecast predicts OBS dry anomaly;) and sum of hit, miss (i.e. forecast misses OBS

dry anomaly), and false (i.e. forecasts predicts dry anomaly whereas OBS don't; CSI= hit/(hit+miss+false)).

### 2.4 The Variable Infiltration Capacity (VIC) model

We used the Variable Infiltration Capacity (VIC, version 4.1.2) (Liang *et al* 1994, 1996) model to simulate hydrologic variables (total runoff and soil moisture) using forcing (daily precipitation, and maximum and minimum temperatures) from

IMD and the forecast products. The VIC model simulates water and energy fluxes at each grid cell and sub-grid variability of precipitation, elevation, soil, and vegetation is well represented (Gao *et al* 2010). The VIC model setup used in this study is well calibrated and evaluated against observed streamflow and satellite based evapotranspiration and soil moisture in Shah and Mishra (2016a) and Shah and Mishra (2016b). The soil and vegetation parameters used in this study are described in Shah and Mishra (2015). The VIC model has been widely used for hydrologic prediction at watershed and regional scales

(Shukla and Lettenmaier 2011, Yuan and Wood 2012b, Mo and Lettenmaier 2014, Shah and Mishra 2016b).

### 2.5 Bias-correction of Precipitation and Temperature Forecast

Improvements in hydrologic prediction were evaluated due to post-processing of meteorological variables (precipitation, maximum and minimum temperatures) from the selected forecast products. We corrected precipitation forecast using the

linear scaling approach as described in Shah and Mishra (2015, 2016b). For each forecast date, we corrected accumulated





precipitation for the selected (7, 15, 30 and 45 days) lead times. We first corrected accumulated precipitation due to extreme events (above 90[th] percentile) for each forecast date in the training period and a scaling factor was obtained for each forecast dates based on ratio of accumulated precipitation for 45 days lead time due to extreme events in the observed and forecast product. In the second step, after the correction for extreme precipitation, scaling factors were obtained based on

accumulated precipitation for 45 days lead time, for each forecast dates from the forecast products and OBS for the entire training period. Scaling factors were estimated for the training period (Nine years), which were evaluated in the testing period (One year). More detailed information on this method can be obtained from Shah and Mishra (2016b).

To correct daily mean (of maximum and minimum) temperature from the forecast, we performed Quantile-Quantile (Q-Q)

mapping (Wood 2002). Initially, we prepared yearly time-series of 45-days lead time average temperature forecast for all the forecast dates along with corresponding observed time-series. For each forecast date and for each grid cell, we estimated quantiles of mean temperature for 45 days lead time for each year using the climatology of the entire period. To estimate quantiles, cumulative distribution functions (CDF) were fitted. Weibull plotting position was used to map cumulative distribution function when percentiles fall between $1/(N+1)$ and $N/(N+1)$; where N is number of climatological years during

the training period. In case when percentiles fall beyond these limits, normal distribution was fitted and values were extrapolated. More details on the Q-Q mapping can be obtained from Shah and Mishra (2016b). Similarly, quantiles were estimated for OBS temperature for corresponding time-series. Based on estimated quantiles, Q-Q mapping was done and forecast was replaced with corresponding value based on OBS. We estimated bias corrected mean temperature using Q-Q mapping. Bias (difference between corrected and uncorrected 45-day average mean temperature) was then added equally to

daily raw Tmax and Tmin to get the corrected values of daily maximum and minimum temperatures. We did not bias correct Tmax and Tmin individually as that will affect the daily temperature range (Tmax-Tmin). We adopted multifold validation approach leaving-one-year out for testing both precipitation and mean temperature (Shah and Mishra 2016b).

Forecast of soil moisture and runoff is essential for planning and decision making in agriculture and water resources. Hence,

we evaluated forecast skill of soil moisture and runoff simulated using meteorological variables from IITM-ensemble (Asoka and Mishra 2015).Using the raw and bias corrected forecasts (precipitation, maximum, and minimum temperatures), the Variable Infiltration Capacity (VIC) model was run to obtain soil moisture and total runoff (surface runoff+baseflow) forecast. We evaluated improvements in correlation of runoff and soil moisture predicted using the bias-corrected precipitation and temperatures from the IITM ensemble (IITM-ensemble-bc) against uncorrected (raw) precipitation and

temperatures from the IITM ensemble mean (IITM-ensemble) and the CFSv2 (Fig S17). For simulating runoff and soil moisture, forcings from all the three products were used to run the VIC model at 0.25º and daily resolution while initial hydrologic conditions were generated using the observed forcing from the IMD. Forecast skill in hydrologic prediction was evaluated for mean total runoff and soil moisture for 7-45 day leads. We considered 45 day lead time to evaluate the





hydrologic prediction skill as for shorter lead times forecast skill are generally higher owing to persistence in initial hydrologic condition.

### 3. Results and discussion

**3.1 Comparison of Forecast Skill for Precipitation and Temperature forecast:**

**3.1.1 Lead time 7 and 15-days**

We estimated forecast skill (against observations, OBS hereafter) in precipitation and air temperature from all the forecast products for 7, 15, 30, and 45 days lead time. Hydrologic forecast at these lead times can be used for planning (field preparation, sowing, irrigation, water management, and reservoir operations) and decision making in water resources and

agriculture. Figure 1shows Spearman Rank correlation between accumulated precipitation forecast and corresponding OBS (IMD) for 7 and 15 days lead times for forecasts initiated during the monsoon months of the period 2001-2009. All the forecast products showed significantly high (more than 0.75) correlation (Fig. 1a-n) in the majority of India for lead time -7 days indicating higher skill for shorter lead time. We noticed that correlation declines as lead time was increased from 7 to 15-days especially in the central region (Fig. 1). Moreover, we find that the GEFSv2 and IITM-ensemble (correlation more

than 0.6 for majority of India) perform better than the CFSv2 for 15 days lead time. Correlations between observed and forecast were generally lower for forecast initiated during the months of July and August (Fig.1o-p). Among all the forecast products, IITM products and their IITM-ensemble mean (mean of all four IITM forecast products) showed better correlations with OBS as compared to the GEFSv2 and CFSv2 for 7 and 15 days lead times (Fig. 1 and Supplementary Table S1). Among the IITM products, products with the atmospheric model operating at higher resolution (IITM-CFST382 and

IITM-GFST382) showed relatively better performance as compared to the other two IITM products, which demonstrates that the models operating at higher resolution provide a better forecast skill (Duffy *et al* 2003, Roebber *et al* 2004).

We estimated MAE in precipitation forecast from all the products as compared to OBS for lead time 7 and 15 days (Fig. S1). We find that MAE is proportional to the magnitude of precipitation as the monsoon season precipitation is higher in the core

monsoon, northeastern, and Western Ghats regions (Fig. S1). Moreover, all the products showed a lower MAE in the arid and semi-arid regions of the western India during the monsoon season and MAE was higher during the months of July-September (Fig. S1o,p). MAE, however, decreases as forecast lead time was increased from 7 to 15 days, which is due to longer accumulation period for precipitation. We noticed that all India median MAE (median of all the grids) in the forecast products vary with the date of forecast, however, both the CFSv2 and IITM ensemble mean showed comparable MAE at all

India scale for 7 day lead (Fig S1o and Table S1). However, for 15 day lead, and for most of the forecast dates (Fig S1p), the IITM-ensemble showed lower error compared to the other products. Overall, based on correlation and MAE, we find that the IITM ensemble performs better than the other forecast products for 7 and 15 day lead time for precipitation prediction.





Lower skill in precipitation forecast in July and August can be attributed to high intraseasonal variability as a large fraction of total precipitation in the monsoon season occurs during these months. Intraseasonal variability can be characterized by spells of active-break periods of length 3-5 days (Rajeevan *et al* 2010). Active-break spells are dominated by SST, wind pattern, Maiden-Julian Oscillation (MJO), and ITCZ (Goswami and Ajayamohan 2000, Woolnough *et al* 2007, Rajeevan *et*

*al* 2010). Predictability of precipitation in India depends on the ability of models to capture intraseaonal and interannual variability in precipitation (Webster *et al* 1998). Improvements in spatial resolution of the atmospheric model and bias corrected SST in the IITM forecast products lead to enhancement in forecast skill, which potentially can be used for decision making in water resources and agriculture in India.

Similar to precipitation for 7 and 15-days lead time, we evaluated skill in maximum (Tmax) and minimum (Tmin) temperatures from all the forecast products against observed air temperatures from IMD (Fig. S2). Tmax averaged for 7 day lead time from all the forecast products showed a good correlation with OBS over the most of India (Fig. S2a-g). Similar to precipitation from the IITM-ensemble, Tmax showed the highest correlation with OBS (0.78; Table S1). However, correlation for 15 days lead time was lower than that of 7 days lead time (Fig S2h-n,p; Table S1). The IITM-ensemble

showed correlation above 0.8 over most of the regions in India and generally skill in Tmax forecast are better than that of precipitation. However, all the forecast products showed a negative correlation (OBS and forecast) in the Northern Himalayan region, which can be partially attributed to sparse gage stations in the complex regions of Himalayas (Mishra 2015).

At 7 days lead time, the forecast products showed higher MAE in the Northwestern arid region, Himalayan range, and Western Ghats (Fig. S3). The IITM products and ensemble mean showed improvement in MAE, which was contributed by enhancements in spatial resolution and bias corrected inputs (SST) in IITM models (Fig. S3a-g,o and Table S1). Overall, the IITM-ensemble showed lower MAE for most of the forecast dates during the monsoon season (Fig S3o and Table S1). Moreover, the IITM-ensemble showed lower all-India median MAE (1.2 ºC) as compared to the GEFSv2 (2.0 ºC) and

CFSv2 (1.7 ºC) for 15 days lead time (Fig. S3h-n,p). Similar to 7 day lead time, all India median MAE in Tmax was the lowest in IITM ensemble for 15 days lead time. CFSv2 models showed better skill in Tmax than GEFSv2, which is consistent with the findings of Shah and Mishra (2016b).

Similar to precipitation and Tmax, forecast skill was estimated based on correlation and MAE for minimum temperature

(Tmin). Tmin from all the forecast products showed lower correlation with OBS as compared to precipitation and Tmax in July-August (Fig. S4). For Tmin, the GEFSv2 (correlations for lead time -7: 0.55 & lead time -15: 0.52) and the IITM-ensemble (0.52 &0.48) showed comparable skill (Table S1). For Tmax and Tmin forecasts, the IITM-ensemble showed lower all-India median MAE as compared to the GEFSv2 (Figs. S3,S5 and Table S1). Predictions of Tmin from all the products showed weaker performance than Tmax, which was also reported in Shah and Mishra (2016b). The difference in





the performance of Tmax and Tmin can be explained as Tmax is mostly governed by partitioning of energy budget  which can be simulated by land surface models) whereas Tmin depends on nighttime boundary conditions and presence of clouds on infrared losses (which may be difficult to simulate) (Pattantyus-Abraham *et al* 2004, Pitman and Perkins 2009). Overall, predictions of Tmin from all the forecast products showed higher errors in the Northwest and Himalayan range and for the most cases, the IITM ensemble outperformed the other forecast products (Fig. S5).

### 3.1.2 Lead time 30 & 45 -days

Since the GEFSv2 reforecast is available only up to lead time 16 days, our comparison for the lead time 30 and 45 days was limited to the forecast products from the IITM and CFSv2. The four IITM products and their ensemble mean showed comparatively better (though not significant) correlations with OBS as compared to the CFSv2 (Figure S6, Table S1). We found that the correlations were higher than 0.5 in the majority of western and central India indicating a reasonable skill at 30 days lead time in the IITM ensemble. However, at 45-days lead, satisfactory forecast skill can only be seen in the arid and semi-arid regions where precipitation amount is substantially lower than the other regions in India (Fig. S6). These results indicate that based on correlations, a reasonable skill can be obtained in the precipitation forecast from the IITM products. Precipitation forecast at lead time 30 and 45 days showed similar spatial patterns of MAE as were observed for the lead time 7 and 15 days (Fig. S7).The IITM ensemble showed an improvement in error over the CFSv2 in the majority of India (Fig. S7). The IITM ensemble mean showed lower error for lead times 30 and 45 days (Fig. S7m,n). This improvement in correlation and MAE can be attributed to finer resolution of the models and bias corrected SSTs, as shown by the IITM-CFST382, and IITM-GFST382 in comparison to IITM-GFST126, IITM-CFST126, GEFSv2, and CFSv2.

Prediction of Tmax from the IITM ensemble showed significant and higher correlation with OBS at 30 days lead time, with major contribution from the IITM-GFST382 product (Fig. S8). We notice that the IITM ensemble showed correlations more than 0.6 for the majority of India between OBS and predicted Tmax at 30 day lead time. At 45 day lead time, correlation decreases (in comparison to 30 day lead time), however, predictions of Tmax from the IITM ensemble mean showed better skill than CFSv2 with OBS. Spatial patterns of MAE in Tmax prediction for lead times 30 and 45 days were consistent with spatial patterns at lead 7 and 15 days indicating larger errors in predicted Tmax in the northern and western parts of the country (Fig S9). Predictions of Tmin showed lower correlation as compared to Tmax (similar to shorter lead times), especially in the Northwest region, where correlations were negative (Fig S10). Predictions of Tmin from the IITM-GFST126 and GFST382 showed better correlation in the southern peninsula. Spatial patterns of MAE in Tmin predictions at lead time 30 and 45 days were consistent with spatial pattern at lead time 7 and 15 days (Fig S11). Predictions from the IITM-CFST382 product showed lower errors as compared to the all other products (Table S1). Predictions of Tmin from the IITM-ensemble mean showed lower error (lead time-30: 0.9 and lead time-45: 1.1 ºC) as compared to the CFSv2 (1.2 and 1.2 ºC) [Table S1]. Overall, the IITM ensemble performs better than the GEFSv2 and CFSv2 for all the lead time (7-45



days). Moreover, the IITM ensemble mean also outperforms other products from the IITM in most of the cases in terms of their individual performance.

Since the IITM ensemble performed better than the other forecast products from IITM, the performance of the IITM

ensemble was compared against the CFSv2 at 7-45dayleads (Fig.2). Since the forecast skill declines with the lead time, we discuss forecast skill at 45 day lead time in details and results for the other leads are presented in supplemental information. At 45 day lead time, correlation in precipitation forecast from the CFSv2 is more than 0.2 only in a few regions (mainly centered in northern and western India) [Fig.2a]. The IITM-ensemble showed correlation (~0.3) higher than the CFSv2 (Fig 2b) in most of region, especially during July-August months (Fig 2c). For Tmax and Tmin forecasts, the IITM-ensemble

showed higher correlations than the CFSv2 in majority of India (Fig 2d,e,g,h).We found that the difference in forecast skill from the IITM-ensemble and CFSv2 is higher at longer lead times. At 7 day lead time, precipitation forecast from the CFSv2 and IITM-ensemble showed correlation more than 0.6 in most regions in India, therefore, at shorter lead times, difference in the forecast skill of CFSv2 and IITM-ensemble is moderate (Fig S12a). For 15 and 30 day lead time, difference in correlations shown by the CFSv2 and IITM ensemble was higher than for 45 days. (Fig S13 and S14). These results show

that the IITM ensemble forecast of precipitation, Tmax, and Tmin have better skill than the CFSv2 for majority of India, which can be used for hydrologic prediction of runoff and soil moisture that can be used for decision making of water resources and agriculture. Moreover, for 30 and 45 day lead time, the IITM-ensemble showed relatively better forecast skill than that of the CFSv2.

**3.2 Performance of bias-corrected IITM-ensemble**

Figure 3 shows improvements in MAE in predictions of precipitation and mean air temperature after the bias correction. Our results showed that the bias correction resulted in reduction in all-India median MAE in precipitation predictions for all the forecast dates during the monsoon season months (Fig. 3c) especially in the Himalayan range and Northeast region (Fig 3a,b). We found substantial improvements in MAE of maximum and minimum temperatures after the bias correction (Figs

3d,e). For instance, all India median MAE was reduced for all the forecast dates after the bias correction (Fig. 3f). Median reduction in MAE for all dates was observed as 2.1 °C. We found that the bias correction substantially improved temperature forecast from the IITM ensemble. This improvement in temperature forecast can be valuable for hydrologic applications. For instance, air temperature influences energy budget in hydrologic models and therefore can affect the partitioning of evapotranspiration and runoff. Due to high intraseasonal variability in the monsoon season precipitation, bias correction

resulted in only marginal improvements in the precipitation forecast.

Fig.S15 shows bias in precipitation forecast from the IITM-ensemble mean, before and after linear scaling for multifold trials using leave-one-year out approach. It can be seen that linear scaling improved negative bias in central India and





Western Ghats and positive bias in the Himalayan range and Southern peninsula. During the testing period (one year), improvement in bias is consistent with the training period (Nine year; Supplementary Fig S15c,d). Improvements in correlation of all-India average precipitation predictions from the IITM-ensemble before and after bias-correction can be noticed (Fig. S16). At 45 days lead time a substantial improvement was noticed as compare to other lead times (Fig. S16d).

Overall, we noticed that the IITM ensemble mean showed improved forecast skill after the bias correction for most of the regions.

### 3.3 Prediction of Soil moisture and total runoff

For all the forecast dates predicted root-zone soil moisture (top 60 cm soil moisture; Fig. S17) showed higher correlation than total runoff (Supplementary Fig S17), which is due to higher persistence in soil moisture as compared to runoff (Shah

and Mishra 2016b). The bias-corrected IITM-ensemble showed higher correlations than the uncorrected IITM-ensemble and the CFSv2. Fig. 4 shows spatial pattern of CSI averaged overall forecast dates for precipitation, daily mean temperature, runoff, and soil moisture for 45 day lead time. CSI of predicting dry anomaly in precipitation using the IITM-ensemble were higher in the Northwestern region whereas lower in Himalayan range and southern peninsula as compared to the CFSv2, which is consistent with the results based on correlation and MAE (Fig. 4). Bias corrected IITM-ensemble showed an

improved CSI in comparison to the raw forecast from the IITM-ensemble and CFSv2 for the majority of regions in India. However, CSI of predicting warm temperature anomalies was lower than that of CSI of predicting dry precipitation anomalies (Fig. 4), especially in Himalayan ranges. This can be on account of higher uncertainty among observations in this region (Mishra 2015). CSI in runoff and soil moisture is higher as compared to precipitation and temperature due to persistence in initial hydrologic conditions (Fig. 4). For 7, 15 and 30 day lead time; CSI is higher than that of 45 day lead

time (Fig S18). We observed that as lead time was increased from 7 to 45 days, CSI of runoff declines in the arid and semi-arid regions of northwest. Overall, we found that the bias correction of forecast improves CSI of precipitation, temperature, total runoff, and soil moisture anomalies in India.

To show the utility of bias corrected forecast in hydrologic prediction in India, we analyzed the forecast for one of the recent

drought years in India. Figure 5 shows anomalies of total runoff and root-zone soil moisture predicted on 15[th]July 2009 for 45 day lead time using the VIC model with the bias corrected IITM-ensemble forecast, which were compared against the observed anomalies (the VIC model was forced with the observed data). Forecast of these hydroclimatic anomalies at sufficient lead time can be helpful in decision making related to water resources and agriculture. We found that the IITM-ensemble-bc successfully captured spatial pattern of observed anomalies, which demonstrates the utility of hydroclimatic

forecast for various applications. Persistence in initial hydrologic conditions simulated using the observed forcing and ability of the IITM-ensemble-bc to capture anomalies in precipitation and temperature (Fig S19) resulted in an improved forecast of total runoff and root-zone soil moisture in the majority of regions in India. However, some overestimation in the areal extent and severity of hydroclimatic anomalies can be noted in central India. These results show that the framework developed





using the IITM-ensemble-bc forecast and the VIC model can be used to predict runoff and soil moisture up to 45 day lead time. Early-warning based on predictions can be helpful in decision making in water resources and agricultural sector so as to minimize risk.

### 4.  Summary and Conclusions

Hydrologic forecast at 7-45 day lead time is essential for decision making in agriculture and water resources. Considering the importance of hydrologic prediction in India, we evaluated the CFSv2, GEFSv2, and forecast products from the IITM. We found that meteorological variables predicted using the IITM products, especially the IITM-ensemble showed better forecast skill than the other two (CFSv2 and GEFSv2) products for all the lead time (7,15, 30, and 45 days) during the monsoon season. We observed improved skill for lead time 30 and 45 days by using the IITM-ensemble in comparison to the

CFSv2, which may be associated with the improvement in model resolution and initial condition used at IITM. For instance, Roxy et al. (2015) reported that the CFSv2 has cold bias of 2-3°C in SSTs which may lead to dry bias in the monsoon season in India. Abhilash et al.(2014a) showed that forcings from GFS and CFS models with bias-corrected SSTs lead to improvement in predictability over the Indian region and that is due to improvement in ability to capture active and break spells. The IITM-ensemble performs better than individual IITM products for most of the selected forecast dates. This is

consistent with findings of Palmer et al.(2004) and Kirtman *et al* (2014), where they reported that multimodel ensemble outperform individual model.

We evaluated the performance of bias corrected forecast from the IITM-ensemble for lead time up to 45 days. Linear-scaling of precipitation forecast and Q-Q mapping of temperature forecast resulted in reduced errors and bias in forecast in India.

Total runoff and root-zone soil moisture forecasts obtained using the corrected IITM-ensemble showed higher skill as compared to the CFSv2 and raw IITM-ensemble for lead time up to 45 days. We found improvements in bias-corrected IITM-ensemble in capturing anomalies of soil moisture and total runoff in comparison to theCFSv2 and raw IITM-ensemble.

Using forcing from the IITM-ensemble and the VIC model, anomalies in precipitation, temperature, root-zone soil moisture, and total runoff were successfully predicted, which can be used in decision making in water resources and agriculture. The bias corrected forecast from the IITM ensemble, which outperforms the GEFSv2 and CFSv2, can be used to develop a hydrologic prediction platform for India. Information on forecast of anomalies in 7-45 day advance with the existing drought monitoring system in India (Shah and Mishra 2015) can be valuable for decision-making in water-resources and agriculture.

The hydrologic prediction based on the IITM-ensemble and the VIC model can provide a basis to predict both meteorological and hydrological anomalies and the information can be provided to farmers and water managers. The forecast



of root-zone soil moisture along with precipitation and temperature anomalies can be used for irrigation planning. Moreover, runoff forecast at 7-45 day lead-time can be valuable for water managers in India.

## Acknowledgement

Authors acknowledge the data availability from the Climate Forecast System (CFSv2), Global Ensemble Forecast System (GEFS), and Indian Institute of Tropical Meteorology (IITM). Financial assistance from the ITRA-Water project was greatly appreciated.

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





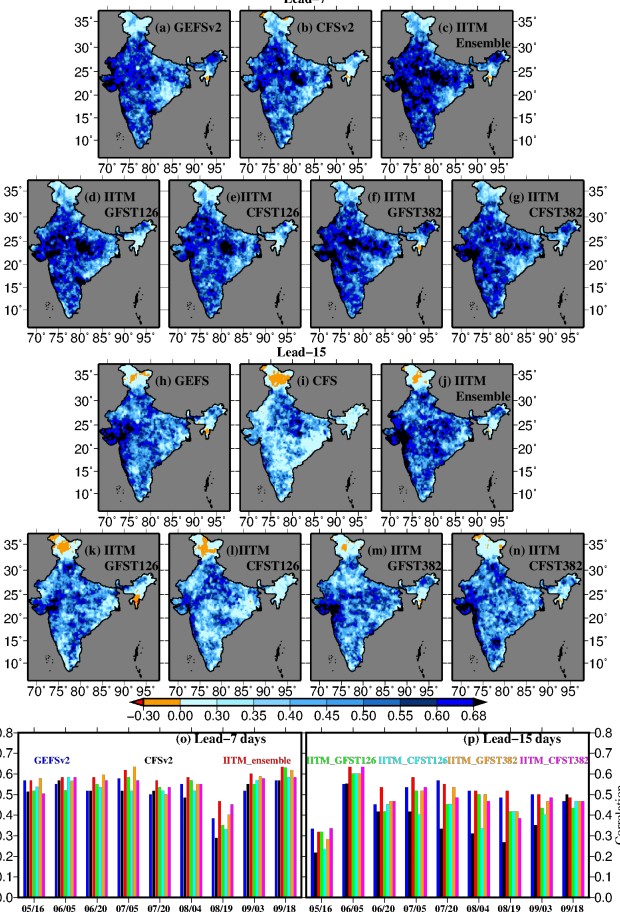

**Figure 1: Correlation between precipitation forecasts and observed precipitation (OBS). (a) Correlation between precipitation forecast from the GEFSv2 accumulated up to 7-days lead time and corresponding OBS, (b) same as (a) but for the CFSv2 (c) same as (a) but for the IITM Ensemble, (d) same as (a) but for the IITM GFST126 (e) same as (a) but for the IITM GFST382, (f) same as (a) but for the IITM CFST382 (h-n) same as (a-f) but for lead time up to 15 days. (o) All-India median correlation between different precipitation forecasts at 7-day lead time and corresponding OBS for the forecasts initiated on different dates (p) same as (o) but for lead time 15 days (period: 2001-2009).**





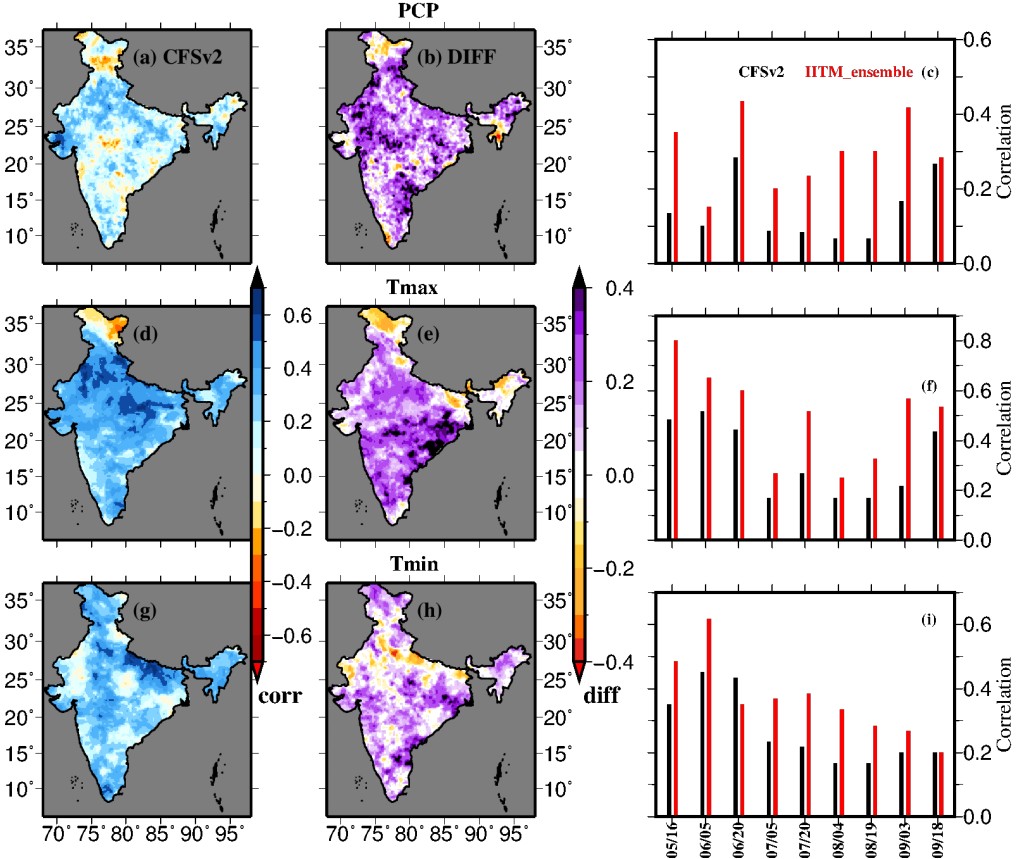

**Figure 2: Improvements in correlations in the IITM-ensemble forecast in comparison to the CFSv2 for 45 day lead time. (a) correlation between precipitation forecast from the CFSv2and OBS (b) change in correlation coefficient of precipitation forecast from the IITM-ensemble and OBS as compared to (a). Correlations in (a) and (b) are median of correlations for the different forecast dates during the monsoon season. (c) All-India averaged median correlation for forecast initiated on different forecast dates. (d-f) is same as (a-c) but for daily maximum temperature and (g-i) is same as (a-c) but for daily minimum temperature.**





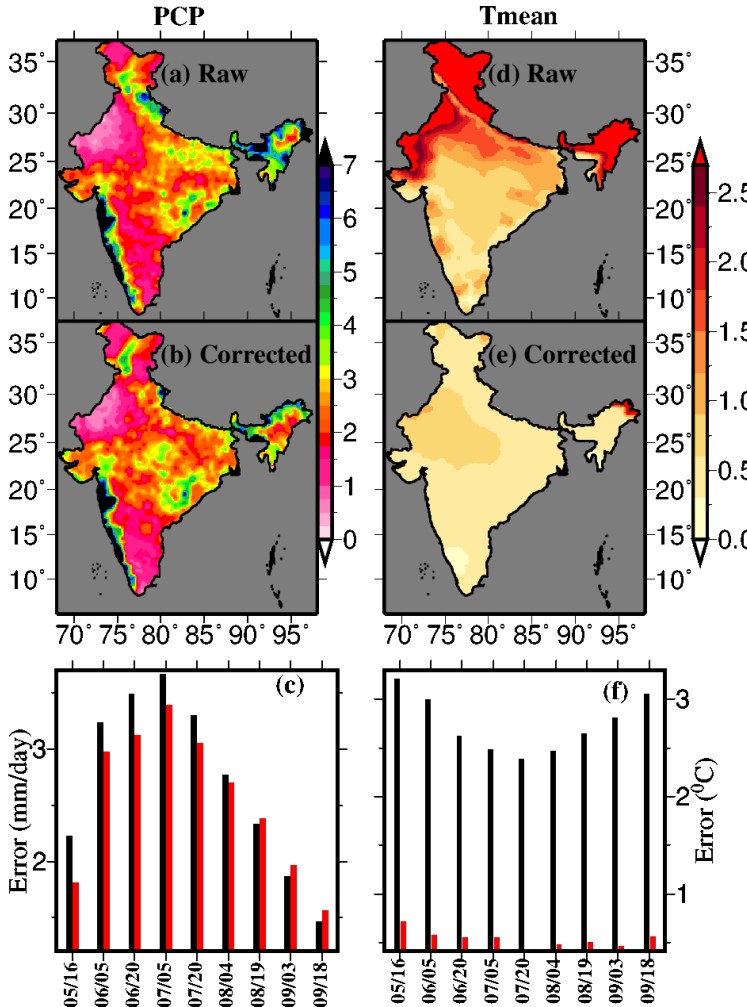

Figure 3: Median absolute error (MAE) in forecast at 45 days lead time from the IITM-ensemble before and after bias correction. (a and b) Median (of all forecast dates) MAE (mm/day) in precipitation forecast before and after bias correction. (c) Comparison of all-India median MAE for each forecast dates (d-f) same as (a-c) but for daily mean temperature in °C.





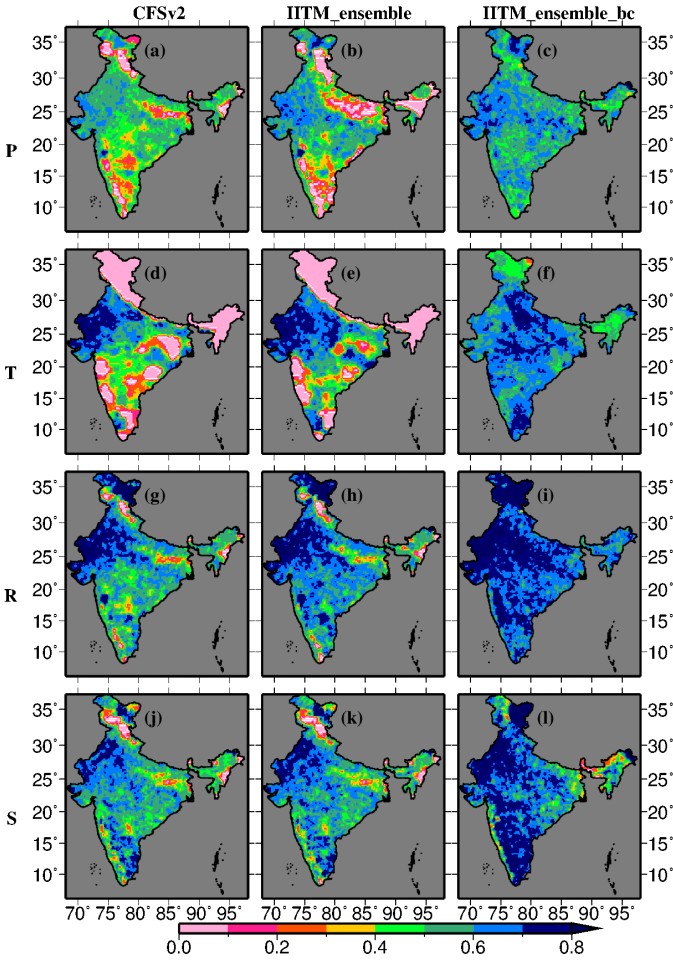

**Figure 4: Critical Success Index (CSI, averaged for forecast dates) of predicting precipitation (a-c), temperature (d-f), runoff (g-i), and soil moisture (j-l) anomalies with respect to the observed anomalies for CSFv2, IITM-ensemble, and bias correctedIITM-ensemble(IITM-ensemble_bc).**





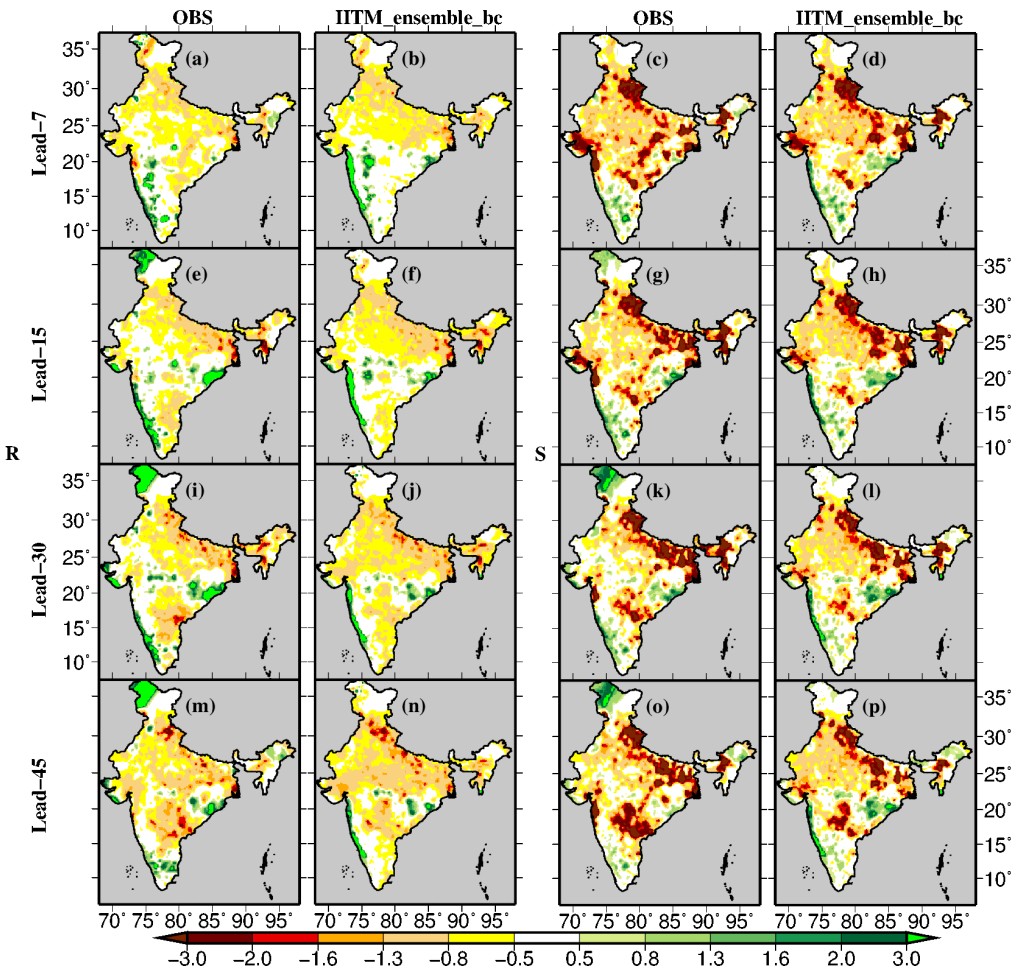

**Figure 5: Predicted anomalies of hydrologic variables for forecast initiated on 15th July, 2009 for lead time 7, 15, 30, and 45 days. (a) Observed (standardized) anomalies in (VIC-simulated) runoff at lead time 7 days (b) anomalies in (VIC-simulated) runoff using bias-corrected IITM-ensemble for lead time 7 days. (c and d) same as (a and b) but for root-zone soil moisture (e-h), (i-l), (m-p) same as (a and b) for lead time 15, 30 and 45 days, respectively.**