# Peer review of "Short to Sub-Seasonal hydrologic forecast to manage water and agricultural resources in India"

_Hydrology and Earth System Sciences, 2016_

## Referee Comment (RC1) · S. Prakash (Referee) · 1 Nov 2016

The manuscript entitled "Short-to-medium range hydrologic forecast to manage water and agricultural resources in India" by R. Shah, A. K. Sahai, and V. Mishra evaluates precipitation and air temperature reforecasts/forecasts from CFSv2, GEFSv2 and IITM models over India for the period of 2001 to 2009. The evaluation is performed at lead times ranging from 7 to 45 days for the southwest monsoon season. The focus of this study is to assess the performance of operational numerical model forecasts for water resources and agricultural practices in India. The topic of research is of broader interest and vital in Indian perspective. The authors have also used bias correction method to precipitation and air temperature forecasts and integrated them in VIC model to assess

total runoff and soil moisture.

Before getting to my specific comments, I hope that my inputs are not taken as criticisms, but as constructive suggestions.

Specific comments: 1. The improvement after bias-correction should be explicitly mentioned quantitatively in "Abstract" and "Conclusion" sections.

2. Please mention the spatial resolutions of the IITM forecast products as well.

3. The spatial resolution of IMD gridded air temperature is 0.5° and also all the model products are available at coarser spatial resolution. But, the assessment is performed at finer spatial resolution of 0.25°. It is suggested to discuss about the propagation of errors due to resampling from coarser to finer spatial resolution with at least one example.

4. The use of mean absolute error (MAE) alone for error quantification might be misleading eventually (Ref: Chai and Draxler, 2014, Geosci. Model Dev., 7, 1247-1250). The use of any normalized error metric would be more appropriate to better understand the error characteristics.

5. Again, CSI is not an equitable categorical metric to evaluate the performance of any numerical model. It is surprising why authors selected CSI alone for this study, even though several better skill metrics are now well-documented.

6. It is suggested to discuss about the impact of sample size at significance of the evaluation in the "Conclusion" section.

7. The authors have appreciably used VIC model here to assess one of the droughts in India. Better prediction of floods is also equally important during the monsoon in India. It would be great if authors demonstrate the same for one flood case too.

8. A careful language check is recommended. For instance, first sentence of page 2 needs to be re-written.

---

## Referee Comment (RC2) · Anonymous Referee #2 · 14 Nov 2016

The analysis contributes to a very important problem in Hydro climatology of Indian subcontinent and provides very useful information towards creating an operational sub-seasonal hydro-meteorological forecasts. The results show a distinct improvement by the IITM forecasts over the NCEP version of CFS 2.0. I have few minor comments, which the authors may address:

1. The authors may highlight, what are the reasons behind such improvements by the IITM model over NCEP CFS v2.0. This should come with some bullet points clearly highlighting the need for any model to be successfully applied for monsoon forecasts.

2. Please, provide some details on the lead-time dependant bias correction. Can this be applied to the CFS2.0 forecasts of precipitation?

[Figure]

3. I could not understand the sources of the observed soil moisture and runoff data. The authors may mention the same or they may provide a table on the details of the data used with their sources. This will help others to reproduce the results and validate the same.

4. During the low rainfall periods, the human intervention is quite high in terms of irrigation. To the best of my knowledge, VIC does not have the capability of doing the same in a way that is applicable to Indian condition. I do not really blame the authors for the same as there is as such no way out, given the status of latest version of VIC. But this should be explicitly mentioned as limitation.

5. Similarly, the crop parameters, which are used in VIC are mostly based on Maize and Soyabean and this is different from Indian crop conditions. The authors may correct me if I am wrong. If I am correct, this should also be mentioned as a limitation. VIC also have limitation of not having a good ground water model. This should also come as a limitation.

6. Is the model calibrated or does it consider the recommended values of parameters of VIC from global data set? The authors may also publish the sensitive parameter values for VIC as supplementary dataset so that the readers will be able to reproduce and apply the work.

---

## Author Response (AR1)

Response to comments

Editor comments

1. The language has been pointed as a weakness of the paper and I strongly urge the authors to perform a thorough spell and grammar check before it can be published. If this is not carried out I will not be able to suggest the paper for publication.

Thanks for your suggestion. We have carefully checked the revised manuscript for grammatical errors.

2. Please provide references to statements on for example farmers committing suicide.

We have remoted the sentence from the revised manuscript as it was based on just media news.

3. P2, L3. Please change "reason" to "driver"

Done

4. You use the term lead time incorrectly. You speak of "7, 15, 30 and 45 days lead times", but you mean accumulation periods or accumulation times. Lead time is the time at which a forecast is compared with observations, the day in which it verifies. If you accumulate a forecast from day 1 to 7 you are get the accumulated forecast for day 7, which strictly speaking is not the same as the lead time of 7 days. I would suggest that you change lead time to accumulation period or accumulation time throughout the manuscript.

Thanks. We have replaced the term 'lead time' with 'accumulation period' as per your suggestion.

5. Generally, sentences like:

"*Fig. 4 shows spatial pattern of CSI averaged overall forecast dates for precipitation, daily mean temperature, runoff, and soil moisture for 45 day lead time.*" and
"*Figure 5 shows anomalies of total runoff and root-zone soil moisture predicted on 15thJuly 2009 for 45 day lead time using the VIC model with the bias corrected IITM-ensemble forecast, which were compared against the observed anomalies (the VIC model was forced with the observed data)*"

are superfluous and should be deleted. This information should be fully contained in the figure caption.

Thanks. We have modified the sentences as suggested.
The manuscript entitled "Short-to-medium range hydrologic forecast to manage water and agricultural resources in India" by R. Shah, A. K. Sahai, and V. Mishra evaluates precipitation and air temperature reforecasts/forecasts from CFSv2, GEFSv2 and IITM models over India for the period of 2001 to 2009. The evaluation is performed at lead times ranging from 7 to 45 days for the southwest monsoon season. The focus of this study is to assess the performance of operational numerical model forecasts for water resources and agricultural practices in India. The topic of research is of broader interest and vital in Indian perspective. The authors have also used bias correction method to precipitation and air temperature forecasts and integrated them in VIC model to assess total runoff and soil moisture. Before getting to my specific comments, I hope that my inputs are not taken as criticisms, but as constructive suggestions.

Thanks. We appreciate your constructive comments and suggestions.

Specific comments: 1. The improvement after bias-correction should be explicitly mentioned quantitatively in "Abstract" and "Conclusion" sections.

Thank you. We have added following text in the Abstract:
 "*Bias corrected precipitation forecast showed an improvement of 2.1 mm (on all-India median MAE) while bias corrected temperature forecast was improved by 2.1°C for 45 days accumulation period*"  on page # 1 in lines#17-18.

The following text has been added in the Conclusions:

"*Bias correction of precipitation and air temperatures resulted an improvement of about 2.1 mm and 2.1°C, respectively in all-India median mean absolute error. Total runoff and root-zone soil moisture forecasts obtained using the corrected IITM-ensemble showed higher skill as compared to the CFSv2 and raw IITM-ensemble for accumulation period up to 45 days. We found that all-India median CSI for runoff forecast was improved from 0.63 to 0.71 after bias correction while CSI of soil moisture forecast was improved from 0.6 to 0.67 for 45 days accumulation period*" on page #13 in lines #23-27.

2. Please mention the spatial resolutions of the IITM forecast products as well.

We have included "*Forecast ensemble members from IITM are available at 1° resolution*" on page# 4 in line # 12

3. The spatial resolution of IMD gridded air temperature is 0.5_ and also all the model products are available at coarser spatial resolution. But, the assessment is performed at finer spatial resolution of 0.25. It is suggested to discuss about the propagation of errors due to resampling from coarser to finer spatial resolution with at least one example.

Thanks. we added the following text (page #4, Lines #29-30):

*"We, however, carefully evaluated all the products at their original spatial resolution and at 0.25° to make sure that datasets are consistent at both resolutions for spatial and temporal variability. We found that the bias in the forecast products at coarser and higher resolution was consistent."*

4. The use of mean absolute error (MAE) alone for error quantification might be misleading eventually (Ref: Chai and Draxler, 2014, Geosci. Model Dev., 7, 1247-1250). The use of any normalized error metric would be more appropriate to better understand the error characteristics.

We compared Normalized RMSE and MAE for lead-7 & 15 days to find errors in different products as shown in figure below. Spatial patterns and overall results obtained from both the matrices were similar. Therefore, we kept MAE in the revised manuscript for further discussion.

[Figure]

Figure 1: Normalized RMSE in precipitation forecast as compared to observed (OBS) precipitation. (a) Error in precipitation forecast accumulated up to 7-days from GEFSv2 as compared to OBS, (b) same as (a) but with CFSv2 (c) same as (a) but with IITM (multimodel, multiresolution) ensemble (d) same as (a) but with IITM GFST126, (e) same as (a) but with IITM CFST126, (f) same as (a) but with IITMCFST382, (g) same as (a) but with IITM GFST382. (h-n) same as (a-g) but for lead 15 days. (o) area-weighted error in different forecast accumulated up to 7-days as compared to OBS for forecast initiated on different monsoon season dates (p) same as (o) but for lead 15 days. (Period: 2001-2009).

5. Again, CSI is not an equitable categorical metric to evaluate the performance of any numerical model. It is surprising why authors selected CSI alone for this study, even though several better skill metrics are now well-documented.

Thanks. We used Equitable Threat score (ETS) as given below where $a_r$ represents number of forecast events captured by chance.

$$ETS = (hit - a_r)/(hit + miss + false - a_r)$$

Where $a_r = (hit + miss)*(hit + false)/n$, n is sample size.

ETS estimated for CFSv2, IITM ensemble, and bias-corrected IITM ensemble mean's performance in capturing dry anomalies is shown in Figure below.

[Figure]

Figure 2 Equitable Threat Score (ETS, averaged for forecast dates) of predicting precipitation (a-c), temperature (d-f), runoff (g-i), and soil moisture (j-l) anomalies with respect to the observed anomalies for CSFv2, IITM-ensemble, and bias corrected IITM-ensemble (IITM-ensemble_bc).

The spatial pattern shown by ETS is similar to that of CSI (Figure 4). This supports our finding that bias-corrected IITM ensemble performs better in capturing dry anomalies. Hence, we used CSI for our analysis.

6. It is suggested to discuss about the impact of sample size at significance of the evaluation in the "Conclusion" section.

We have discussed the influence of sample size in the revised manuscript and added the following text:

 "*One of the limitations of evaluation of the products in this study is small sample size. The evaluation of all the forecast products was based on 10 common years and 9 forecast dates during the monsoon season. Increasing the sample size in future based on the availability of forecasts for longer period may further improve evaluation and the bias correction.*" on page #13 lines #9-11.

7. The authors have appreciably used VIC model here to assess one of the droughts in India. Better prediction of floods is also equally important during the monsoon in India. It would be great if authors demonstrate the same for one flood case too.

The focus of the present study was to evaluate hydrologic prediction for drought assessment (page #2, line#18-20), which has been mentioned in the last paragraph of Introduction. Assessment of floods is an important topic, which requires significant work related to observed data collection and development of robust routing models. Therefore, that will be considered in a separate manuscript.

8. A careful language check is recommended. For instance, first sentence of page 2 needs to be re-written. Interactive comment on Hydrol. Earth Syst. Sci. Discuss., doi:10.5194/hess-2016-504, 2016

The manuscript has been carefully checked for possible errors related to grammar.

Hydrol. Earth Syst. Sci. Discuss.,
doi:10.5194/hess-2016-504-RC2, 2016

**Interactive comment on "Short-to-medium range
hydrologic forecast to manage water and
agricultural resources in India" by Reepal Shah
et al.**

**Anonymous Referee #2**

The analysis contributes to a very important problem in Hydro climatology of Indian
subcontinent and provides very useful information towards creating an operational subseasonal
hydro-meteorological forecasts. The results show a distinct improvement by
the IITM forecasts over the NCEP version of CFS 2.0. I have few minor comments,
which the authors may address:

1. The authors may highlight, what are the reasons behind such improvements by the
IITM model over NCEP CFS v2.0. This should come with some bullet points clearly
highlighting the need for any model to be successfully applied for monsoon forecasts.

We appreciate your suggestion and added the following text in the revised manuscript (page #13, lines #14-18)

*"The major factors that might have contributed in the improvements in the IITM forecast are:*
  *i. Ensemble members of IITM forecast are generated by perturbing initial atmospheric conditions to improve simulation of the northward propagation*

  *ii. Improvements in the boundary conditions with bias corrected SST result in an improved precipitation prediction*

  *iii. Higher spatial resolution of the IITM forecast can better resolve orographic rainfall"*

2. Please, provide some details on the lead-time dependant bias correction. Can this
be applied to the CFS2.0 forecasts of precipitation?

Thanks for your valuable suggestion. We have provided following (page #11 Lines #25-28)
*"However, the bias in the forecast products may have temporal variability and may not be constant for the
entire period of 45 days. Therefore, bias correction approaches based on the variable lead time (Stockdale,
1997) need to be evaluated in future when IITM forecast for long-term retrospective period is available. The
bias correction approach that we presented can be applied to evaluate seasonal forecast skill."*

3. I could not understand the sources of the observed soil moisture and runoff data.
The authors may mention the same or they may provide a table on the details of the
data used with their sources. This will help others to reproduce the results and validate
the same.

Thanks. The VIC model was calibrated and evaluated using observed streamflow and satellite soil moisture and
evapotranspiration (Shah and Mishra, 2016a,b). In this study, we used calibrated VIC model forced with
observed IMD data to simulate soil moisture and runoff, which was considered as a reference to evaluate the
forecast of soil moisture and runoff.

We have provided following (page#11 in lines#30-35)

*"The VIC model was calibrated and evaluated using observed streamflow and satellite soil moisture and
evapotranspiration (Shah and Mishra, 2016a and Shah and Mishra, 2016b).In this study, we used calibrated
VIC model forced with observed IMD data to simulate soil moisture and runoff, which was considered as a
reference to evaluate the forecast of soil moisture and runoff. Forecast of root-zone soil moisture and runoff
was simulated using the VIC model forced with the forecast products (IITM-ensemble-bc, IITM-ensemble, and
CFSv2), which were evaluated against the soil moisture/runoff obtained from the VIC model simulation using*

*the observed forcing from IMD (Supplemental Fig. S17)."*

4. During the low rainfall periods, the human intervention is quite high in terms of irrigation. To the best of my knowledge, VIC does not have the capability of doing the same in a way that is applicable to Indian condition. I do not really blame the authors for the same as there is as such no way out, given the status of latest version of VIC. But this should be explicitly mentioned as limitation.

We appreciate your valuable comment. The present study considers only the monsoon season for analysis, however, we acknowledge that the role of irrigation during the dry season.

We have mentioned this limitation and added the following text (page #6 Lines 8-11) in the revised manuscript:

*"The VIC model's version that was used in this study does not explicitly represent groundwater, rather it only accounts for baseflow. We acknowledge that India specific soil and vegetation parameters along with the representation of irrigation, reservoir, and groundwater can improve the water budget; however, these were not considered in the present study due to unavailability of either observations or the model version that has the representation of human interventions."*

5. Similarly, the crop parameters, which are used in VIC are mostly based on Maize and Soyabean and this is different from Indian crop conditions. The authors may correct me if I am wrong. If I am correct, this should also be mentioned as a limitation. VIC also have limitation of not having a good ground water model. This should also come as a limitation.

Thanks. The point is valid.

We included the following text to address this (page #6, Line 4-8):

*"The vegetation parameters used in this study were developed using 1-km Advanced Very High Resolution Radiometer (AVHRR) global land cover information. We used vegetation library that was developed at University of Washington. The vegetation parameters were not specifically developed to incorporate crops that are grown in India. However, the existing parameters were successfully used in the model application over India (Shah and Mishra, 2015; Shah and Mishra, 2016)."*

6. Is the model calibrated or does it consider the recommended values of parameters of VIC from global data set? The authors may also publish the sensitive parameter values for VIC as supplementary dataset so that the readers will be able to reproduce and apply the work.

We appreciate your suggestion. However, the model calibration and evaluation is a part of the previously published manuscript (Shah and Mishra (2015, 2016) and Shah and Mishra (2016)), therefore, we are unable to provide the calibrated values of parameters in this study. Authors will be happy to share the parameters and data, if someone is interested.

We have mentioned following text in the revised manuscript (page #6 lines 11-13):

[revised manuscript text omitted]

~~air temperature using Q-Q mapping and estimated the bias between corrected and uncorrected mean temperature, which was added to Tmax and Tmin to obtain daily maximum and minimum temperatures. Since the sample size was small, we evaluated the effectiveness of the Q-Q mapping using pooled data for all the selected forecast dates and found that results were consistent with corrected temperature for specific forecast dates. Figure 3 shows improvements in MAE in predictions of precipitation and mean air temperature after the bias correction. Our results showedfoundfound~~find that the bias correction substantially improved temperature forecast from the IITM ensemble. This improvement in temperature forecast can be valuable for hydrologic applications. For instance, air temperature influences energy budget in hydrologic models and therefore can affect the partitioning of evapotranspiration and runoff. Due to high intraseasonal variability in the monsoon season precipitation, bias correction resulted in only marginal improvements in the precipitation forecast.

We find that linear scaling improved negative bias in precipitation forecast in central India and Western Ghats and positive bias in the Himalayan range and Southern peninsula. During the testing period (one year), improvement in bias is consistent with the training period (Nine year;  Fig S15c,d). Improvements in correlation of all-India average precipitation predictions from the IITM-ensemble before and after bias-correction can be noticed (Fig. S16). At 45 days accumulation period a substantial improvement was noticed as compare to other accumulation periods (Fig. S16d). Overall, we noticed that the IITM ensemble mean showed improved forecast skill after the bias correction for most of the regions.We bias corrected the forecast products for the accumulation period of 45 days. However, the bias in the forecast products may have temporal variability and may not be constant for the entire period of 45 days. Therefore, bias correction approaches based on the variable lead time (Stockdale, 1997) need to be evaluated in future when IITM forecast for long-term retrospective period is available. However, the bias correction approach that we presented can be applied to evaluate seasonal forecast skill

**3.3 Prediction of Soil moisture and total runoff**

The VIC model was calibrated and evaluated using observed streamflow and satellite soil moisture and evapotranspiration (Shah and Mishra,2016a and Shah and Mishra,2016b). In this study, we used calibrated VIC model

forced with observed IMD data to simulate soil moisture and runoff ~~simulated using meteorological variables from IITM ensemble (Asoka and Mishra 2015).Using the raw and bias corrected forecasts (precipitation, maximum, and minimum temperatures), the Variable Infiltration Capacity (VIC) model was run to obtain soil moisture and total runoff (surface runoff+baseflow) forecast. We evaluated improvements in correlation of runoff and soil moisture predicted using the bias corrected precipitation and temperatures from the IITM ensemble (IITM-ensemble-bc) against uncorrected (raw) precipitation and temperatures from the IITM ensemble mean (IITM ensemble) and the CFSv2 (Fig S17). For simulating runoff and soil moisture, forcings from all the three products were used to run the VIC model at 0.25° and daily resolution while initial hydrologic conditions were generated using the observed forcing from the IMD. Forecast skills in hydrologic prediction were evaluated for mean total runoff and soil moisture for 7-45 day leads. We45 day lead timehydrologic prediction skills as for shorter lead times forecast skills are generally higher owing to persistence in initial hydrologic condition.~~forecast of soil moisture and runoff. Forecast of root-zone soil moisture and runoff was simulated using the VIC model forced with the forecast products (IITM-ensemble-bc, IITM-ensemble, and CFSv2), which were evaluated against the soil moisture/runoff obtained from the VIC model simulation using the observed forcing from IMD (Fig. S17). For all the forecast dates predicted root-zone soil moisture (top 60 cm soil moisture; Fig. S14) showed higher correlation than total runoff ( Fig S17), which is due to higher persistence in soil moisture as compared to runoff  (Shah and Mishra, 2016b). The bias-corrected IITM-ensemble showed higher correlations  than the uncorrected IITM-ensemble and  CFSv2.

~~We evaluated forecast skill of the IITM-ensemble for hydroclimatic anomalies using Critical Success Index (CSI). For estimation of CSI, models ability to predict dry anomaly were considered for precipitation, runoff, and soil moisture (i.e. standardized anomaly < 0) and warm anomaly for air temperature (i.e. temperature anomaly >0). Observed climatology was used to estimate anomalies of precipitation, air temperature, soil moisture and runoff. 
[revised manuscript text omitted]

---

## Editor Decision (ED1)

Editor comments

1. The language has been pointed as a weakness of the paper and I strongly urge the authors to perform a thorough spell and grammar check before it can be published. If this is not carried out I will not be able to suggest the paper for publication.

2. Please provide references to statements on for example farmers committing suicide.

3. P2, L3. Please change "reason" to "driver"

4. You use the term lead time incorrectly. You speak of "7, 15, 30 and 45 days lead times", but you mean accumulation periods or accumulation times. Lead time is the time at which a forecast is compared with observations, the day in which it verifies. If you accumulate a forecast from day 1 to 7 you are get the accumulated forecast for day 7, which strictly speaking is not the same as the lead time of 7 days. I would suggest that you change lead time to accumulation period or accumulation time throughout the manuscript.

5. Generally, sentences like:

"*Fig. 4 shows spatial pattern of CSI averaged overall forecast dates for precipitation, daily mean temperature, runoff, and soil moisture for 45 day lead time.*"  and
 "*Figure 5 shows anomalies of total runoff and root-zone soil moisture predicted on 15thJuly 2009 for 45 day lead time using the VIC model with the bias corrected IITM-ensemble forecast, which were compared against the observed anomalies (the VIC model was forced with the observed data)*"

are superfluous and should be deleted. This information should be fully contained in the figure caption.